# Cost analysis of treating cardiovascular diseases in a super-specialty hospital

Atul Kumar[1], Vijaydeep Siddharth[1]*, Soubam Iboyaima Singh[1‡], Rajiv Narang[2‡]

1 Department of Hospital Administration, All India Institute of Medical Sciences (AIIMS), New Delhi, India,
2 Department of Cardiology, Cardio-Thoracic Centre, AIIMS, New Delhi, India

☯ These authors contributed equally to this work.
‡ These authors also contributed equally to this work.
* dr.siddharthmamc@gmail.com

**Data Availability Statement:** All relevant data is within the paper.

**Funding:** The authors received no specific funding for this work.

## Abstract

Cardiovascular care is expensive; hence, economic evaluation is required to estimate resources being consumed and to ensure their optimal utilization. There is dearth of data regarding cost analysis of treating various diseases including cardiac diseases from developing countries. The study aimed to analyze resource consumption in treating cardio-vascular disease patients in a super-specialty hospital. An observational and descriptive study was carried out from April 2017 to June 2018 in the Department of Cardiology, Cardio-Thoracic (CT) Centre of All India Institute of Medical Sciences, New Delhi, India. As per World Health Organization, common cardiovascular diseases i.e. Coronary Artery Disease (CAD), Rheumatic Heart Disease (RHD), Cardiomyopathy, Congenital heart diseases, Cardiac Arrhythmias etc. were considered for cost analysis. Medical records of 100 admitted patients (Ward & Cardiac Care Unit) of cardiovascular diseases were studied till discharge and number of patient records for a particular CVD was identified using prevalence-based ratio of admitted CVD patient data. Traditional Costing and Time Driven Activity Based Costing (TDABC) methods were used for cost computation. Per bed per day cost incurred by the hospital for admitted patients in Cardiac Care Unit, adult and pediatric cardiology ward was calculated to be Indian Rupee (INR) 28,144 (US$ 434), INR 22,210 (US$ 342) and INR 18,774 (US$ 289), respectively. Inpatient cost constituted almost 70% of the total cost and equipment cost accounted for more than 50% of the inpatient cost followed by human resource cost (28%). Per patient cost of treating any CVD was computed to be INR 2,47,822 (US $ 3842). Cost of treating Rheumatic Heart Disease was the highest among all CVDs followed by Cardiomyopathy and other CVDs. Cost of treating cardiovascular diseases in India is less than what has been reported in developed countries. Findings of this study would aid policy makers considering recent radical changes and massive policy reforms ushered in by the Government of India in healthcare delivery.

## Introduction

Globally an estimated 17.7 million people died from cardiovascular diseases (CVDs) in 2015 representing 31% of all deaths and over 75% of these deaths take place in low and middle

**Competing interests:** None declared.

income countries [1]. India is having the highest burden of CVDs in the world and as per World Health Organization (WHO) estimates Non Communicable Diseases (NCDs) accounts for 60% of total deaths in India out of which CVD tops the list of with 43.34%. [2]. A study estimating the burden of cardiovascular diseases in India revealed that from 1990 to 2016 prevalent cases of cardiovascular diseases increased from 25·7 to 54.5 million [3].

Increasing NCDs coupled with increasing injuries have resulted in a significant increase in health spending in India causing loss to national income [4]. CVDs not only impact the well-being of an individual but also holds back the economic growth of the country due to increased healthcare expenditure and diminished productivity from disability, premature death, and absenteeism. WHO has estimated that India has spent approximately $236 billion over a period of 10 years between 2005–2015 on management of CVDs [5–7]. It imposes a considerable burden on regions with relatively low per capita health budgets [8].

Rapid scientific and technological advancement has tremendously increased the cost of managing CVDs. Individually, patients with CVDs incur more than twice the medical costs as compared to a patient without CVD of same age and sex [9]. Cardiovascular care is expensive; hence, economic evaluation is required to estimate resources being consumed and to ensure their optimal utilization [10]. It will facilitate rational allocation of resources and informed decision making to formulate effective policies [11].

The information obtained from cost analysis helps the healthcare institutions to operate more cost effectively, monitor and control costs, improve quality of care, and aid in management decision making, particularly cost containment by making costing profile of procedures more accurate [12]. There is dearth of data regarding cost analysis of treating various diseases including cardiac diseases from developing countries. Thus, the present study is an attempt to analyze the cost of managing admitted cardio-vascular disease patients in a super-specialty hospital.

## Methodology

An observational and descriptive study was carried out from April 2017 to June 2018 in the Department of Cardiology, Cardio-Thoracic Centre (CTC) of All India Institute of Medical Sciences, New Delhi, India. All India Institute of Medical Sciences is one of the leading healthcare institute in the country and CTC began functioning in 1982 with 200 hospital beds for Cardiology and Cardiothoracic & Vascular Surgery patients. It has all the facilities required for comprehensive cardiac care including a catherization laboratory, cardiac anesthesia, cardiac radiology, cardiac biochemistry, cardiac pathology, nuclear medicine, stem cell facility, organ retrieval & banking organization, blood transfusion services etc. It has four general wards, 8 Operating rooms, two intensive care units having 34 beds, eight ICU beds exclusively for the neonatal & infant intensive care, five cardiac catheterisation laboratory and one coronary care unit (CCU). CTC runs various super-specialty clinics (Coronary Clinic, Prosthetic valve Clinic, Hypertension Clinic, Arrhythmia/ Pacemaker Clinic, Heart Failure Clinic, Aortic disease Clinic).

This study was conducted as part of post graduate residency programme in the Department of Hospital Administration after obtaining necessary approval from the Institute Ethics Committee for Post Graduate Research vide IECPG-685/19.01.2017, RT31/16.02.2017. Study did not involve any direct or indirect interaction with patients and ethical guidelines as prescribed by the Institute Ethics Committee for the patient record review were adhered. It entailed identifying various cost centres, classifying costs and tracing all costs related to treating cardiovascular diseases through detailed and thorough perusal of various records including inpatient records to ascertain resource consumption during hospitalisation.

CVDs such as Coronary Artery Disease (CAD), Rheumatic Heart Disease (RHD), Cardio-myopathy, Congenital heart diseases, Cardiac Arrhythmias etc. were included in the study as classified by WHO and as per the expert guidance [13]. Since, the study was carried out as part of partial fulfilment of MD programme in hospital administration, only 100 patient care records were studied and number of patient records for a particular CVD was identified using prevalence-based ratio of admitted CVD patient data. Medical records were studied for the specific entire admission episode. Medical records of the patients who were admitted for either less than 24 hours or admitted through emergency department were excluded from the study due to operational issues of data collection and resource constraints. As per the Institute policy, approval from the Hospital Administration is required for studying medical records of the admitted patients for research or academic purpose.

'Process mapping' of various patient care services during the hospital stay was conducted. It was done through direct observations and in-depth discussions with the key informants i.e. faculty members, resident doctors and nurses etc. Since the availability of data was a challenge due to lack of robustness in record keeping practices, combination of methodology utilizing using Traditional Costing and Time Driven Activity Based Costing (TDABC) were used for cost computation. Replacement method of cost computation employing Cost Inflation Index (CII—As notified under the Finance Act) was adopted for arriving at the current day cost from the historical costs of various capital assets (CII in the year 2010–11 was 167, while for the year 2017–18, it was 280, hence, CII factorial increase was calculated to be 1.89). Annualised cost was calculated for various cost centres and subsequently per bed per day cost was arrived Table 1.

## Capital costs

**Building and its maintenance cost.** Measurement of the patient care areas (in sqm) were taken from the engineering department for the purpose of calculation of construction cost as per Central Public Works Department (CPWD) Manual 2007 (CII multiplication index 2.29). It was estimated that life of the building would be 100 years and therefore, an annual depreciation of 1% would be a reasonable estimation of the annual cost of the building. As per repair & service cost index as on 24/04/2018 issued by office of Director General, CPWD regulation, the maintenance rate for engineering works was Rs 6668.56/- per Sq.m. per year for the hospital building. No cost has been attributed to the land since the land belongs to the Government of India.

**Cost of equipment, fixtures, and its maintenance cost.** A list of equipment used in the Cath lab, CCU and inpatient wards was populated, and its procurement cost was collected from hospital stores section. Straight-line method of depreciation (useful life 10/ & 7 years) was used to arrive at an annualized equipment cost. As per the prevailing practices five percent of the total procurement cost of an equipment was taken as maintenance cost and same was added to the annualized equipment cost to arrive at the total cost of equipment or an asset. For catheterization lab procedures, activity-based costing was utilized, while for CCU and wards, traditional method of costing was used.

## Operational costs

**Human resource.** Monthly gross salary was computed for various categories of staff. Time spent by various categories of staff during Cath lab procedures was decided in consultation with key informants (Consultants, Residents and Nursing Officers), thereafter, apportioning was done. While, in CCU and ward, the gross monthly salary of different manpower under various categories was computed by considering the actual number of manpower multiplied by the gross salary in that category.

**Table 1. Various cost centres included in the study.**

| CAPITAL COSTS | OPERATIONAL COSTS |
|---|---|
| **Cost of Equipment** | **Salary** |
| **Cost of Building** | Faculty |
| **Overhead Cost** | Residents |
| Electricity | Nursing Staff |
| Laundry services | Technical staff |
| Air-conditioning | Hospital Attendant |
| Central Sterile Supply Department (CSSD) | Sanitary Attendant |
| Manifold services | Security Guard |
| Bio-Medical waste management | **Overhead Costs** |
| Dietary services | Electricity |
| Manifold services | Laundry services |
| Bio-Medical waste management | Air-conditioning |
| Dietary services | CSSD |
| Manifold services | Manifold services |
| | Bio-Medical waste management |
| | Dietary services |
| | Maintenance cost |
| | **Hospital supplies** |
| | Surgical store |
| | Medical store |
| | General store |
| | Linen store |
| | Stationary store |
| | **Investigation** |
| | Laboratory Investigation |
| | Radiological Investigation |

**Medicines, surgical and other consumables.** Treatment file of each enrolled patient was studied for ascertaining consumption of carious consumables. Consumption pattern of various items i.e. general items, surgical items, stationary, linen store etc. over a period of six months was generated from hospital MIS database of the respective areas. Unit cost of each item was taken from the hospital stores and cost of various items consumed over a period of six months was calculated.

**Air conditioning & electricity costs.** Cost for total tonnage of refrigeration (TR), Air handling unit (AHUs), condenser and chiller pumps, cooling tower was taken from engineering services division. Operational cost was taken based on its tonnage of refrigeration and electricity consumed per day including labor, spares, and material. Wattage and usage of various electrical appliances and fixtures was ascertained. Total number of units consumed in 24 hours by all the appliances and fixtures was calculated and total cost was arrived at.

**Support services costs.** Reference costs for support services was taken from the various studies carried out by the Department of Hospital Administration at AIIMS, New Delhi. Total per day load of dirty linen (in kg) of various categories generated from patient care areas under study was calculated with the help of Nursing Officer In charges and laundry supervisor. Cost of washing per kilogram of dirty linen was taken as Rs 23.22. Cost of serving one meal in the general ward was estimated to be Rs. 86.77, while unit cost of breakfast was taken as 15% of the cost of a meal. Cost of sterilizing items was estimated to be Rs. 33.9 per bed per day. The number of total medical gas outlets for each patient care area was calculated and Rs 25.56 was

**Table 2. Hospital admissions of cardiovascular diseases and study sample.**

| S. No | Disease Group | Total Cardiology Admissions (%) | Study sample (n = 100) |
|---|---|---|---|
| 1 | Coronary artery diseases | 4756 (55.65%) | 30% |
| 2 | Rheumatic heart disease | 887 (10.38%) | 24% |
| 3 | Congenital Heart Disease | 638 (7.47%) | 14% |
| 4 | Cardiac Arrhythmias | 637 (7.45%) | 7% |
| 5 | Cardiomyopathy | 576 (6.74%) | 14% |
| 6 | Others | 1052 (12.31%) | 11% |

taken as cost per manifold point [14]. Cost for managing biomedical waste management per day per bed was calculated. Additional 10% cost was added to the overall cost under the administrative head. All costs were calculated in INR and US $ (1US $ = 64.8 INR)

## Results

Cardiology department provides services to patients from various parts of the country and neighboring countries as well. It has an OPD of 42000 patients, 9500 inpatient admissions per year (Average length of stay of 6.6 days), and more than 7000 cardiac interventions (More than 30 different types of cardiac procedures) were carried out in the year 2016–17. Most common CVD was CAD, which constituted more than 50 percent of the admitted patients. Table 2 A total of 100 admitted patients were enrolled in the study for calculating cost of various kinds of cardiac illnesses Table 3. Patient care flow has been depicted in Fig 1.

Costs were computed distinctly for the Cardiac Care Unit (CCU), adult and pediatric ward owing to variation equipment being used, manpower allocation, and consumption of various consumables. Median length of stay in case of CCU, adult and pediatric ward was observed to be one day (1–9 days), five days (1–26 days) and six days (2–57 days), respectively. Cost per bed per day incurred by the hospital for admitted patients in CCU, adult and pediatric

**Table 3. Socio demographic profile of patients in the study (n = 100).**

| S No. | Parameters | Subgroup | Number (n = 100) |
|---|---|---|---|
| 1 | Gender | Female | 35 |
| | | Male | 65 |
| 2 | Age | 0–18 | 22 |
| | | 19–40 | 22 |
| | | 41–60 | 36 |
| | | >60 | 20 |
| 3 | Region | UP | 29 |
| | | Bihar | 28 |
| | | Delhi | 23 |
| | | Haryana | 14 |
| | | Other states | 6 |
| 4 | Socioeconomic Status (According to modified Kuppuswamy scale 2017) | Upper | 4 |
| | | Upper Middle | 33 |
| | | Lower Middle | 42 |
| | | Upper Lower | 19 |
| | | Lower | 2 |

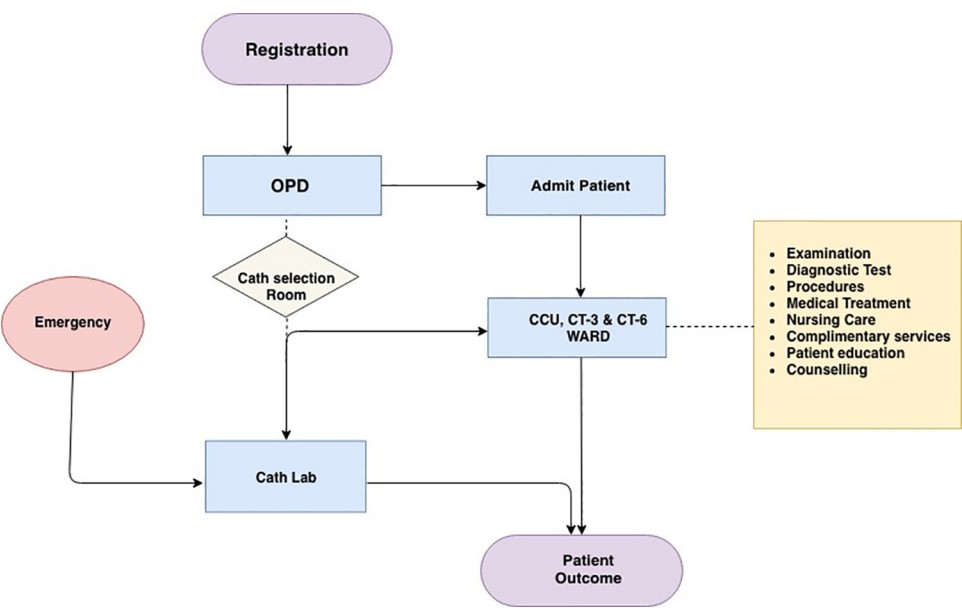

**Fig 1. Patient care flow at Cardio-Thoracic Centre.**

cardiology ward was calculated to be INR 28,144 (US$ 434), INR 22,210 (US$ 342) and INR 18,774 (US$ 289), respectively. Overall, per patient cost of managing any cardiovascular illness in hospital was computed to be INR 2,25,293 (US $ 3476) and cost after including administrative overheads was estimated to be INR 2,47,822 (US $ 3824) Table 4.

Equipment cost accounted for the maximum expenditure followed by human resource cost, both together accounted for more than 70 percent of the per bed per day cost. After analyzing results of the direct costs, it was observed that more than 65% of the cost is attributable to hospitalization. Interventional/diagnostic procedures cost is more than 18% while medication treatment represented only 1.5% of the total direct costs Fig 2.

Congenital heart disease is the most expensive to be treated followed by rheumatic heart disease, cardiomyopathy and so on Table 5. Treatment cost of RHD is on a higher side because of the increased length of stay. In cases of cardiomyopathies, cost of treatment was higher because of the costly intra cardiac devices.

## Discussion

A study done in India found that prevalence of CVDs increased in all states of India, and coronary artery diseases tops the list among CVDs followed by stroke, while rheumatic heart

**Table 4. Cost of care to the hospital for treating cardiovascular diseases patients.**

| | Cost of care in INR (US$) | Percentage |
|---|---|---|
| Inpatient cost [Per day cost = INR 16991 (ALOS = 10 days)] | 169,910 (2,622) | 69 |
| Procedure's cost | 46,491 (717) | 19 |
| Laboratory Investigations | 4,633 (71) | 2 |
| Medications Cost | 3,459 (53) | 1 |
| Radiological Investigations | 800 (12) | 0 |
| Administrative overheads | 22,529 (347) | 9 |
| **Per patient cost of treating CVD illness** | **247,822 (3,824)** | 100% |

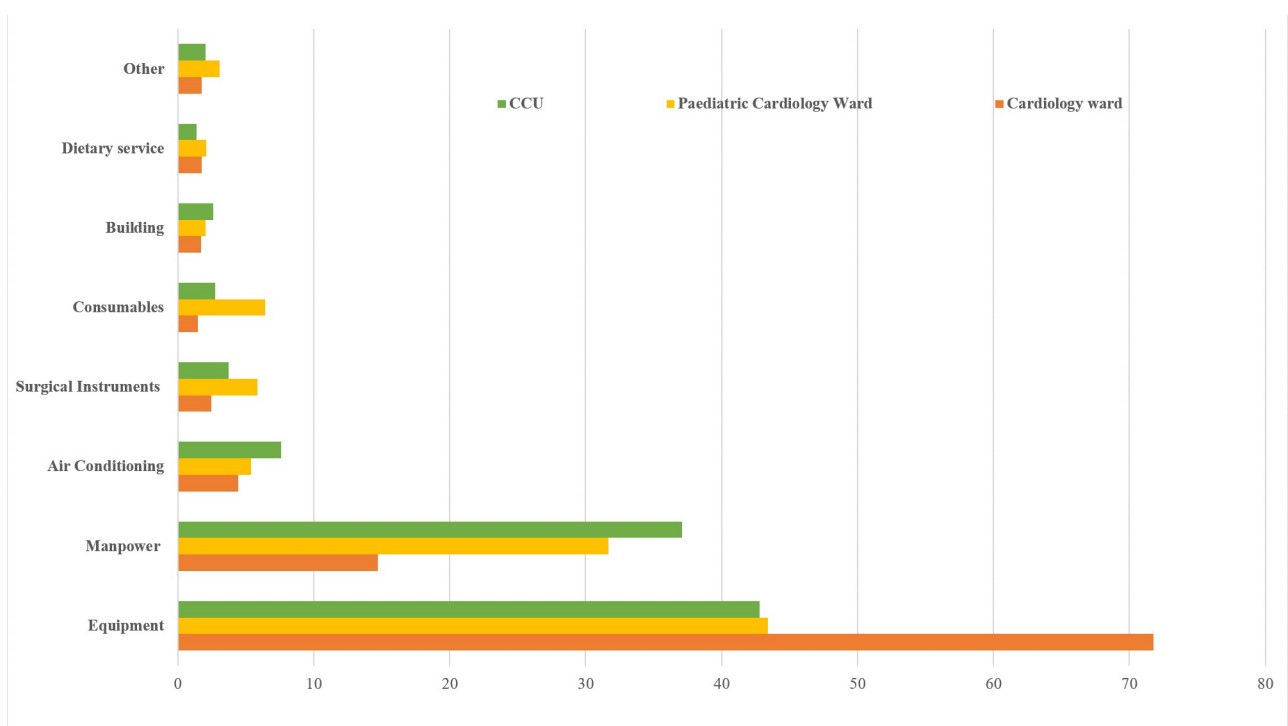

**Fig 2. Various cost centers of inpatient cost in percentage (per bed per day cost).**

disease decreased by 10·8% from 1990 to 2016, and similar findings have been observed in the current study [15]. The cost of patient care is galloping with the advancement of technology and cost of consumables. Hence, it is realized that the cost of patient care along with the required procedures during the hospitalization should be known to policymakers to have standardized approach and devise packages or revise existing packages, for various CVDs.

The findings of this cost analysis will be helpful in health planning and in rational allocation of finite resources especially in developing countries. With rapid transition of Indian Healthcare delivery system from predominantly out of pocket backed based to assurance based with public sector funding, these estimates may prove to be particularly useful in coming up with package cost for treating various CVDs under Ayushman Bharat–Pradhan Mantri Jan Arogya Yojana (Flagship scheme of Govt of India for providing cashless treatment to more than 500 million beneficiaries through public and private sector hospitals) [16].

**Table 5. Cost to hospital for treating various cardiovascular illnesses in INR(US$).**

| Diseases | Median Length of stay | Treatment cost | Inpatient Cost | Medication Cost | Radiological Investigation Cost | Laboratory Investigation Cost |
|---|---|---|---|---|---|---|
| **Coronary Artery Disease** | 2 | 157,840 (2,435) | 96,025 (1,481) | 2,595 (40) | 219 (3) | 3,157 (48) |
| **Rheumatic Heart Disease** | 11 | 298,759 (4,610) | 249,411 (3,848) | 5,297 (81) | 380 (5) | 7,061 (108) |
| **Congenital Heart Disease** | 6 | 311,583 (4,808) | 231,855 (3,578) | 2,036 (31) | 740 (11) | 4,830 (74) |
| **Cardiomyopathy** | 6 | 273,042 (4,213) | 154,037 (2,377) | 630 (9) | 1,753 (27) | 4,534 (69) |
| **Other cardiovascular illnesses** | 5 | 260,722 (4,023) | 151,208 (2,333) | 5,752 (88) | 1,630 (25) | 3,778 (58) |

Cost analysis of cardiac diseases is extremely significant, because of it being the leading cause of morbidity and mortality. However, cost analysis in a developing country is fraught with major challenge of data availability which leaves scope for much to be desired [17–19]. In this study, the comprehensive cost i.e. both direct and indirect cost (Building and equipment cost) of treating CVD patients in a hospital setting for the year 2017–18 was estimated to be US $ 3476. A systematic review on the economic burden of cardiovascular disease and hypertension in low- and middle-income countries, included eighty-three studies of which 50% were from China, Brazil, India and Mexico. Most of the studies were single center retrospective cost studies conducted in secondary care settings. The costs per episode for hypertension and generic CVD were fairly homogeneous across studies; ranging between $500 and $1500. In contrast, for coronary heart disease (CHD) and stroke cost estimates were generally higher and more heterogeneous, with several estimates in excess of $5000 per episode [20].

According to a study conducted in USA, the total mean direct medical care costs for patients with established cardiovascular disease (CVD) was $18,953 per patient per year with inpatient costs being 42.8% ($8114) of total costs [21]. While, cost per CVD hospitalization in 2012 in Sanghai, averaged US $ 2236.29 with the highest being for chronic rheumatic heart diseases (US $ 4710.78) [22]. Cost of treating CVDs in developed countries appears to be much higher compared to what has been observed in this study. A study carried out in Brazil, the cost of hospitalizations for heart failure and myocardial infarction was estimated to be R$ 3,085.15 per patients which is lower, when compared to this study [23]. Findings of our study is similar to a study which reported a total direct cost of €3198 per patient (1 Euro = 81.69), with largest part of the expenses (79%) attributable to hospitalization (ward), while laboratory investigations and medical treatment accounted for 17% and 4%, respectively [24].

In China, a retrospective study was conducted to ascertain the direct medical costs among 10000 inpatients of coronary heart diseases. It revealed that the average hospitalization expenses were $6791.38 and top three expenses were medical consumables, procedure charges and drugs [25]. A study conducted in Iran in year 2016, the average total cost per patient was observed to be US$1881, with hospitalization cost as the major cost center [26]. A study on assessment of the direct cost of treatment of ischemic heart disease from a tertiary care hospital of Pakistan, revealed mean total cost of care to be Pakistan Rupee (PKR) 3,59,975.00. Majority of the cost expenses were contributed by the procedure cost (Rs 2,73,574), followed by laboratory and diagnostic cost (Rs. 37684); hospital stay cost (Rs. 27,697) and medication cost (Rs 21,019), which is similar to the present study [27]. Similar findings were observed in a study carried out in Iran where highest level of expenditure under direct medical costs were observed on angiography, hospitalization and drug supply [28].

Findings of this study can be generalized to a similar setting in a developing country especially South East Asia Region. Individuals in Low- and Middle-Income countries bear significant financial burdens following CVD hospitalization, yet with substantial variation across and within countries. Lack of insurance may drive much of the financial stress of CVD in LMIC patients and their families [29]. Health coverage and financial risk protection and inequality in access to health care remains a serious issue for South Asian countries. Greater progress is needed to improve treatment and preventive services and financial security. Findings of this study can provide guidance for developing packages for implementing insurance programme or reimbursement schemes through govt sponsored schemes [30]. Non communicable diseases (NCDs) impose a substantial financial burden on many households, including the poor in low-income countries. The financial costs of obtaining care also impose insurmountable barriers to access for some people, which illustrates the urgency of improving financial risk protection in health in LMIC settings and ensuring that NCDs are taken into account in these systems [31].

Presented study has comprehensively considered almost all the cost centers (both direct & indirect) attributing to the expenses in management of CVDs including capital and machinery costs, while most of the studies focus only on direct costs, which sets it apart from other studies and is the major strength of the present study. However, due to limited availability of data sources and other resource constraint activity-based cost analysis could not be carried. Most cases of suspected CVDs across the country are referred to tertiary care institutes for diagnosis and treatment, with this study being a hospital-based study, it is possible that referral bias could have played a role. The study sample was limited to patients admitted through the Outpatient Department of Cardiology and did not include the emergency admissions, day care cases and those patients who required surgical treatment under Cardio-Thoracic Vascular Surgery (CTVS). Similar costing studies, focusing on both direct and indirect costs involved in the management of cardiovascular diseases with larger sample size should be carried out to increase the generalizability of the findings. It is also equally important to measure economic burden in terms of informal care and loss of productivity, which contributes to half of the economic burden of CVDs.

## Conclusion

Cardiovascular disease is a major public health problem in India and is associated with high economic burden. Coronary artery disease is the most prevalent disease among the CVDs. Cost of treating Rheumatic Heart Disease is the highest among all CVDs followed by Cardio-myopathy and other CVDs. Cost of treatment in CVDs in developed countries is reported to be higher compared to what has been observed in this study. The results of the study would be valuable to health policy makers considering recent radical changes and large-scale policy reforms ushered in by the Government of India in healthcare delivery system.

## Author Contributions

**Conceptualization:** Atul Kumar, Vijaydeep Siddharth, Soubam Iboyaima Singh, Rajiv Narang.

**Data curation:** Atul Kumar.

**Formal analysis:** Atul Kumar, Vijaydeep Siddharth.

**Methodology:** Atul Kumar, Vijaydeep Siddharth, Soubam Iboyaima Singh, Rajiv Narang.

**Project administration:** Rajiv Narang.

**Resources:** Soubam Iboyaima Singh, Rajiv Narang.

**Supervision:** Vijaydeep Siddharth, Soubam Iboyaima Singh, Rajiv Narang.

**Validation:** Vijaydeep Siddharth.

**Writing – original draft:** Atul Kumar, Vijaydeep Siddharth.

**Writing – review & editing:** Atul Kumar, Vijaydeep Siddharth, Soubam Iboyaima Singh, Rajiv Narang.

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
