## [Decision Letter · Decision Letter 0]

4 Aug 2021

PONE-D-21-18894

Cost analysis of treating cardiovascular diseases in a super-specialty hospital

PLOS ONE

Dear Dr.Vijaydeep Siddharth

Thank you for submitting your manuscript to PLOS ONE. After careful consideration, we feel that it has merit but does not fully meet PLOS ONE’s publication criteria as it currently stands. Therefore, we invite you to submit a revised version of the manuscript that addresses the points raised during the review process.

ACADEMIC EDITOR:

The article is interesting because it explores a very important topic in patient management. However, to be publishable on PLOSONE it is necessary that the Authors answer the following points:

1. To respond accurately to the Reviewers’ requests

2. To improve the methodology of the proposed study

3. To highlight how the results of the study can be valid in contexts other than the one where the study was carried out

4. To improve the bibliography by adding more recent studies on the subject, particularly researches carried out in Developing Countries other than India.

5. To explain all the acronym used (CTVS, SAARC, INR, PKR) and in particular:

In the abstract the acronym CVDs at line 41.

Line 86 and 95-96 : clarify the observed sample and the criteria chosen to define the sample

Line 102 add the acronym CII after Cost Inflation Index

In table 1 explain the acronym CSSD.

Ther decision is justified on PLOS ONE’s publication criteria.  

Please submit your revised manuscript by August 31, 2021.  If you will need more time than this to complete your revisions, please reply to this message or contact the journal office at plosone@plos.org. Please include the following items when submitting your revised manuscript:

We look forward to receiving your revised manuscript.

Kind regards,

Filomena Pietrantonio

Academic Editor

PLOS ONE

Journal Requirements:

2. Please provide additional details regarding participant consent. In the ethics statement in the Methods and online submission information, please ensure that you have specified whether consent was written or verbal/oral. If consent was verbal/oral, please specify: 1) whether the ethics committee approved the verbal/oral consent procedure, 2) why written consent could not be obtained, and 3) how verbal/oral consent was recorded. If your study included minors, please state whether you obtained consent from parents or guardians in these cases. If the need for consent was waived by the ethics committee, please include this information

3. In your Methods section, please provide a justification for the sample size used in your study, including any relevant power calculations (if applicable)

4.Thank you for stating the following financial disclosure: 

  "unfunded"

5. We note that you have referenced (ie. Bewick et al. [5]) which has currently not yet been accepted for publication. Please remove this from your References and amend this to state in the body of your manuscript: (ie “Bewick et al. [Unpublished]”) as detailed online in our guide for authors

Additional Editor Comments (if provided):

The article is interesting because it explores a very important topic in patient management. However, to be publishable on PLOSONE it is necessary that the Authors answer the following points:

1. To respond accurately to the Reviewers’ requests

2. To improve the methodology of the proposed study

3. To highlight how the results of the study can be valid in contexts other than the one where the study was carried out

4. To improve the bibliography by adding more recent studies on the subject, particularly researches carried out in Developing Countries other than India.

5. To explain all the acronym used (CTVS, SAARC, INR, PKR) and in particular:

In the abstract the acronym CVDs at line 41.

Line 86 and 95-96 : clarify the observed sample and the criteria chosen to define the sample

Line 102 add the acronym CII after Cost Inflation Index

In table 1 explain the acronym CSSD

Reviewers' comments:

Reviewer's Responses to Questions

**Comments to the Author**

1. Is the manuscript technically sound, and do the data support the conclusions?

Reviewer #1: Partly

Reviewer #2: Yes

2. Has the statistical analysis been performed appropriately and rigorously? 

Reviewer #1: No

Reviewer #2: No

3. Have the authors made all data underlying the findings in their manuscript fully available?

Reviewer #1: Yes

Reviewer #2: Yes

4. Is the manuscript presented in an intelligible fashion and written in standard English?

Reviewer #1: Yes

Reviewer #2: Yes

5. Review Comments to the Author

Reviewer #1: The article needs substantial changes. In order to estimate costs of different procedures an activity based costing approach should be adopted. This means that you should split clearly in the manuscript the three different steps for cost estimation:

- identification of resources;

- measurement;

- valorization

as per the identification step, you should show flow charts depicting the entire process of take-in - charge, from admission to discharge. Moreover, you should identify clearly where and when each activity takes plaace. I would suggest to focus only on one or two rpocedures within the cardiovascular domain.

Finally, you shoulsd assess the variability and the generalizability of your results to other context. A montecarlo simulation would be very useful in this case.

Reviewer #2: The paper explores and gives data on the total costs for CVD in developing countries.

The study is based on a retrospective observational design, it is mostly sound and adequately discussed, and I believe it could be accepted for pubblication after some minor issues are solved.

Here a list of suggestions for the Authors:

1 - No reporting guideline has been explicitely used in the study. Unfortunately, journal policies state that articles should adhere to appropriate reporting guidelines and community standards for data availability. I suggest to identify a guideline of choice (i.e. STROBE) and follow it in the reporting.

- METHODOLOGY

2 - L.86: It is claimed that "Study did not involve any patients" - however, it is stated in several instances that a sample of 100 patients was taken for data modeling. Authors should clarify the meaning of this sentence. Some ambiguity is present in other section of the paper aswell (see L95-96 "no involvement of patients or public, vs. L 234 "The study sample was limited to patients admitted...")

3 - L. 87: It is claimed that "ethical guidelines as deemed appropriate for this study were adhered". Please specify the guidelines the Authors are referring to.

4 - L. 93-94: "A total of 100 admitted patients of various CVDs were enrolled in the study using prevalence-based sampling and were followed up till discharge". This approach of taking data from a sample in order to draw conclusions should be accompanied by more rigorous definition of sample size calculation. Authors should clarify the reason that exact number of patients was deemed necessary and suffiecent for correct study conduction.

5 - L 94-95: Please give reasoning for exclusion criteria

- RESULTS

6 - Since data from the sample is taken in order to draw more general conclusions on procedures costs and to compare this against other literature results, it would be more appropriate to clarify the uncertainty level in the study, by indicating confidence intervals for calculated values, and - if the authors deem it feasible - by performing some form of sensitivity analysis.

- DECLARATIONS

7 - It is claimed that "Consent to participate was taken for studying medical records". However, no statement of patient involvement is declared.

6. PLOS authors have the option to publish the peer review history of their article (what does this mean?). If published, this will include your full peer review and any attached files.

Reviewer #1: No

Reviewer #2: **Yes: **Antonio Vinci

---

## [Decision Letter · Decision Letter 1]

20 Dec 2021

Cost analysis of treating cardiovascular diseases in a super-specialty hospital

PONE-D-21-18894R1

Dear Dr. Vijaydeep Siddharth,

We’re pleased to inform you that your manuscript has been judged scientifically suitable for publication and will be formally accepted for publication once it meets all outstanding technical requirements.

Kind regards,

Filomena Pietrantonio

Academic Editor

PLOS ONE

Additional Editor Comments:

The Authors addressed all comments in a satisfactory way. The manuscript is now suitable for publication.

Reviewers' comments:

Reviewer's Responses to Questions

**Comments to the Author**

1. If the authors have adequately addressed your comments raised in a previous round of review and you feel that this manuscript is now acceptable for publication, you may indicate that here to bypass the “Comments to the Author” section, enter your conflict of interest statement in the “Confidential to Editor” section, and submit your "Accept" recommendation.

Reviewer #1: All comments have been addressed

Reviewer #2: All comments have been addressed

2. Is the manuscript technically sound, and do the data support the conclusions?

Reviewer #1: Yes

Reviewer #2: Yes

3. Has the statistical analysis been performed appropriately and rigorously? 

Reviewer #1: Yes

Reviewer #2: Yes

4. Have the authors made all data underlying the findings in their manuscript fully available?

Reviewer #1: Yes

Reviewer #2: Yes

5. Is the manuscript presented in an intelligible fashion and written in standard English?

Reviewer #1: (No Response)

Reviewer #2: Yes

6. Review Comments to the Author

Reviewer #1: all revisions have been addressed properly abd accurately and the manuscript is now suitable for publication

Reviewer #2: The Authors addressed all comments in a satisfactory way. I believe the paper can be accepted for publishing.

7. PLOS authors have the option to publish the peer review history of their article (what does this mean?). If published, this will include your full peer review and any attached files.

Reviewer #1: No

Reviewer #2: **Yes: **Antonio Vinci

---

## [Editor Report · Acceptance letter]

27 Dec 2021

PONE-D-21-18894R1 

Cost analysis of treating cardiovascular diseases in a super-specialty hospital 

Dear Dr. Siddharth:

I'm pleased to inform you that your manuscript has been deemed suitable for publication in PLOS ONE. Congratulations! Your manuscript is now with our production department. 

Kind regards, 

on behalf of

Dr. Filomena Pietrantonio 

Academic Editor

PLOS ONE